# The Transition to Noncommunicable Disease: How to Reduce Its Unsustainable Global Burden by Increasing Cognitive Access to Health Self-Management

**DOI:** 10.3390/jintelligence9040061

**Published:** 2021-12-09

**Authors:** Linda S. Gottfredson

**Affiliations:** School of Education, University of Delaware, Newark, DE 19716, USA; gottfred@udel.edu

**Keywords:** intelligence, functional literacy, job complexity, nonadherence to treatment, noncommunicable disease, diabetes, diabetes self-management, behavioral risk factors, global burden of disease, epidemiological transition

## Abstract

The global epidemic of noncommunicable diseases (NCDs), such as cardiovascular disease and diabetes, is creating unsustainable burdens on health systems worldwide. NCDs are treatable but not curable. They are less amenable to top-down prevention and control than are the infectious diseases now in retreat. NCDs are mostly preventable, but only individuals themselves have the power to prevent and manage the diseases to which the enticements of modernity and rising prosperity have made them so susceptible (e.g., tobacco, fat-salt-carbohydrate laden food products). Rates of nonadherence to healthcare regimens for controlling NCDs are high, despite the predictable long-term ravages of not self-managing an NCD effectively. I use international data on adult functional literacy to show why the cognitive demands of today’s NCD self-management (NCD-SM) regimens invite nonadherence, especially among individuals of below-average or declining cognitive capacity. I then describe ways to improve the cognitive accessibility of NCD-SM regimens, where required, so that more patients are better able and motivated to self-manage and less likely to err in life-threatening ways. For the healthcare professions, I list tools they can develop and deploy to increase patients’ cognitive access to NCD-SM. Epidemiologists could identify more WHO “best buy” interventions to slow or reverse the world’s “slow-motion disaster” of NCDs were they to add two neglected variables when modeling the rising burdens of disease. The neglected two are both cognitive: the distribution of cognitive capacity levels of people in a population and the cognitive complexity of their health environments.

## 1. Modern Life Is Becoming Ever More Cognitively Complex

In his essay, “The Evolution of Idiots,” humorist Scott [1] ([1]) describes how “a few thousand amazingly smart deviants” created a world that turned the rest of us into ninnies. We were pretty much doomed, he says, after “some deviant went and built the printing press…. Civilization exploded. Technology was born. The complexity of life increased geometrically. Everything got bigger and better. Except our brains”.

Adams captures perhaps the most pervasive but least appreciated influence that general intelligence, the latent trait (phenotypic *g*), has on modern life: the recurring wide variation in *g* within human populations gradually reshapes a culture as advances in technology ratchet up the cognitive complexity of everyday life ([18]). In so doing, successive innovations make differences in cognitive ability more salient. They begin to “ninnify” most everyone, but especially individuals of lower or declining capacity, because they are less able to capitalize on their culture’s new benefits and avoid its new hazards.

My specific concern is that, with advances in treatment regimens for noncommunicable diseases (NCDs), health care providers are unknowingly placing cognitive burdens on patients too heavy for many to bear. Cognitive overload precludes adherence to NCD treatments, which patients themselves must implement to stave off predictable complications and premature death. Healthcare providers are vexed and public health officials alarmed by high rates of nonadherence. The major NCDs—heart disease, stroke, cancer, diabetes, and chronic respiratory disease—are epidemic worldwide, their morbidity and mortality steadily rising, and their costs becoming unsustainable for rich and poor countries alike: a global “slow-motion disaster”, warned the WHO ([12]). I use adult literacy data to explain why NCDs are so difficult to self-manage and suggest tools that the healthcare professions can develop and deploy to increase patients’ cognitive access to NCD self-management (NCD-SM). I also suggest how epidemiologists can incorporate cognitive risk in their models to identify “best buys” ([42]) for controlling the global burdens of NCDs.

## 2. Noncommunicable Diseases (NCDs) Now Cause More Disease, Disability, and Death Globally Than Do Infectious Diseases

Figure 1 shows the remarkable global progress in eliminating the pestilences that killed people en masse in earlier millennia, including malaria, yellow fever, smallpox, and cholera. Deaths from communicable diseases fell swiftly after global immunization and clean-water campaigns were launched in poorer countries in the late 20th century, as they had fallen in richer countries earlier in the century ([9]). During 2000–2019 alone, the crude death rate (CDR) was almost halved in countries the World Bank classified as low income in 2019 and cut by 25% in the lower-middle income country group (Panels a, b), both their CDRs dropping well below that of aging high-income countries. Life expectancy at birth increased in all country groups, but increased most—by over 11 and 7 years—in the two lowest income groups as infant mortality and deaths from diarrheal diseases plummeted (WHO GHO database, https://www.who.int/data/gho/data/themes/mortality-and-global-health-estimates/ghe-leading-causes-of-death (accessed on 5 December 2021)).

Infectious diseases are still major killers in the poorest nations and new ones periodically flame into pandemics, but there will soon be fewer deaths from infectious disease than NCDs in low-income countries too ([30]). NCDs are diseases of modernity and rising prosperity. They are increasing in all country income groups, and rates of NCD prevalence, morbidity, and mortality in developing nations are converging on those of the richest ([16]). As the epidemiological transition to NCDs becomes global, so do its challenges.

## 3. NCDs Are Harder for Governments to Prevent and Control Than Infectious Diseases

Communicable diseases are caused by biological agents not visible to the naked eye: bacteria, viruses, protozoa, fungi, and parasitic worms. They incubate unseen for days to weeks before symptoms appear and, upon arrival, tend to be acute and obvious (diarrhea, vomiting, chills, high fever, respiratory distress, skin lesions, swelling, pain, body rash). Throughout history such scourges decimated cities and armies with frightening speed. Public health and medical technologies dramatically reduced their ravages worldwide in little over a century by reducing involuntary exposure to the pathogens (e.g., safer water, sewer, and food security systems; vector control), reducing the susceptibility of individuals to infection if exposed to the pathogen (e.g., immunization), and improving survivability among infected individuals (antibiotics). These are primarily top-down interventions to stop community transmission of infectious agents.

These strategies do not work with NCDs. The disease process depicted in Table 1 shows why. They are not caused by involuntary exposure to biological agents, but mainly by voluntary self-exposure to known health-damaging products and lifestyles: using tobacco, eating an unhealthy diet, misusing alcohol, and being sedentary. These behaviors might seem innocuous in the short-run but, if habitual, they progressively damage multiple organ systems. Their population-attributable fractions, summed, show they were responsible for 35.2% of all global deaths in 2019.

These health-damaging habits are easy to acquire but hard to break because they are evolutionarily novel perks of modernity and growing affluence. Like Trojan horses, we welcome their dangers into our lives as pleasures and conveniences. Alcohol and tobacco are physically addictive. The industrially manufactured ultra-processed food products that now dominate our diets ([41]) are seductive, if not addictive, because they are designed to override evolved satiety signals ([23]). As the advertisement for a new potato chip in the 1980s boasted: “Betcha can’t eat just one!” Other modern conveniences and enticements such as the automobile, television, video games, and social media lure us into sitting for long hours. Indeed, schools require it, as seen in the high rates of physical inactivity—79–85%—among school-going youth in all four World Bank income groups ([22]). The epidemiological transition to NCDs was wrought by this transition in health risks.

As it mounts, the internal damage these behaviors cause arouses no alarm because it produces no obvious symptoms. Health screenings can reveal the tell-tale signs of system damage—high blood pressure, blood glucose levels, BMI, and LDL cholesterol—but these conditions do not produce any outwardly noticeable symptoms of disease either. Together, this nexus of metabolic risk factors accounts for 47.3% of deaths worldwide. 

A recent global study of 87 risk factors concluded that “there has been no real progress (since 1990) reducing exposure to behavioural risks, while metabolic risks are, on average, increasing every year (by 1.37%/year)” ([16]). Of particular concern is the steep increase in obesity among children and adults, at least doubling in all four country income groups since 1980 and tripling in the high-income group ([30]). Large prospective studies find that obesity can explain about 50% of diabetes type 2[note 1] cases ([14]), and diabetes is a risk factor for other NCDs, especially coronary heart disease and chronic kidney disease ([4]). The NCDs listed in Table 1 were responsible for 39.9% of all global deaths in 2019. 

NCDs are easy to ignore even after diagnosis. Taking oral medications to lower high blood pressure, blood sugar, and cholesterol provides no discernible benefit day to day, nor does failing to take them seem to do harm. Many patients therefore fail to take them as prescribed ([6]). And so it is with overeating and underactivity too. Even when the risk of complications later in life is acknowledged, they are too distant to seem urgent. Only the arrival of NCDs’ predictable complications is unmistakable: heart attack, stroke, kidney failure, limb amputation, blindness, peripheral neuropathy, among others.

Table 2 points to a troubling trend noted in another global study ([15]). That study used seven metrics to measure the global burden of disease 1990–2019: number of cases and age-standardized rates of prevalence, incidence, death, years lost to premature death (YLLs), years of healthy life lost to disability (YLDs), and disability-adjusted years of life (DALYs, the sum of YLLs and YLDs). The concern is that public health has focused primarily on life-saving interventions for NCDs, such as heart attack, stroke, and lung cancer, which have reduced their fatality. But there has been much less on interventions for less fatal but more disabling NCDs, such diabetes and COPD. The former three NCDs accounted for 78% of global deaths attributed to the leading five NCDs in 2019, but only 31% of all their cases and 31% of their healthy years lost that year to living with a disability (YLDs).

As populations continue to age, the prevalence of disabling NCDs keeps rising, but their burden in YLDs will increase disproportionately. This will severely tax healthcare systems worldwide. The effect of diabetes will be disproportionate, because it has by far the largest YLD rate, and its prevalence is rising fast in countries of all income levels (WHO https://www.who.int/health-topics/diabetes (accessed on 5 December 2021)). This is additionally concerning because public health campaigns have had limited success in preventing obesity, diabetes’ biggest risk factor, or in gaining adherence to treatment among people diagnosed with diabetes ([29]). 

## 4. International Surveys of Adult Functional Literacy Point to a Common Fundamental Cause of Nonadherence to NCD Treatments: The Cognitive Complexity of NCD Self-Management

Health literacy is “the degree to which individuals have the capacity to obtain, process, and understand basic health information needed to make appropriate health decisions” (https://www.hrsa.gov/about/organization/bureaus/ohe/health-literacy/index.html (accessed on 5 December 2021). The U.S. Surgeon General ([5]) explained why it is critical: “As clinicians, what we say does not matter unless our patients are able to understand the information we give them well enough to use it to make good health-care decisions. Otherwise, we didn’t reach them, and that is the same as if we didn’t treat them.” She also drew attention to 75% of Americans having trouble taking their medications as directed ([6]). Seeking the cause, [28] ([28]) identified specific sets of behaviors and barriers. All were cognitive: inability to perceive relevance, stay vigilant to what is relevant, weigh pros and cons, and imagine unobservable benefits; insufficient information processing capacity for the complexity of medication management; and inaccurate, irrational, or conflicting beliefs about medications. None of these cognitive skills are specific to medication. All are generalizable. 

The international surveys of adult literacy listed in Figure 2 confirm that literacy is a capacity for processing information ([33]). It is not just a collection of content-specific KSAs (knowledge, skills, and abilities), but the ability to acquire and use them effectively. They also reconfirmed what work-literacy researchers proved long ago ([40]). Literacy is not merely the ability to decode written words, but to understand what you read to successfully complete a given task. It is not reading to know, but to do; to accomplish the myriad tasks of daily life or on a job with what you read, hear, observe, learn, know, or can figure out—essentially, fluid intelligence. Not surprisingly, the separate scales on each literacy survey intercorrelate highly and line up along a single general factor ([24], Tables 12-2 and 12-3), as do the various subtest scores on intelligence test batteries ([8]).

Figure 2 shows the population-level distribution of literacy proficiency scores grouped into 5 broad levels. In all four surveys about 50% of the population in these high-income countries is proficient at literacy levels 1 or 2, but no higher, and by age 80 virtually everyone functions at these lowest two levels. 

Figure 3 lists sample items for each and graphs error rates for people of different proficiency levels performing tasks at different difficulty levels. People who peak at a given level have an 80% chance of correctly performing (20% of failing) tasks at that difficulty level. For instance, people who peak at a proficiency level 2 have a 20% chance of failing level 2 items, such as finding two specified pieces of information in a short sports article. Their likelihood of failing level 1 items falls to 4% but they have, at best, a 50/50 chance on level 3 tasks. This includes the vast majority of individuals aged 70 and older (Figure 2, Panel d). Conversely, the minority of people who routinely function at proficiency levels 4 or 5 will make few errors on level 1–3 tasks and err at low rates on the more difficult level 4–5 tasks.

Especially important, the surveys’ developers determined why the more difficult items (those failed more often) were more cognitively demanding. It was not content area (e.g., health, finances), format (prose, graphs, forms), or readability per se ([24]). It was the complexity of the information processing—mental manipulations—they required for people to answer correctly: more abstract concepts, more distant inferences, more plausible distractors, and more information to integrate. 

To illustrate, the sample tasks at literacy level 2 require locating two specific pieces of information but the level 1 tasks only one piece. The items at level 3 require figuring out where to place a given piece of information in a two-variable matrix or calculating a result from two clearly specified variables in a chart. The level 4 items require finding a pattern along a timeline or the similarities and differences between two benefits plans based on descriptions of them, which requires independently recognizing which information is most relevant, then analyzing it to answer a specific question. The level 5 items require the same independent recognition and analysis of relevant information, though in denser text, to draw a conclusion, but they also require evaluating that conclusion against a self-generated standard. Appendix A uses a nutrition label to illustrate other specific contributors to task complexity.

The 3-D landscape of probable error in Figure 3 shows how differences in people’s abilities and tasks’ difficulty levels predict population error rates like clockwork when work tasks are instrumental and performed independently without assistance. Lower ability and higher complexity both increase risk of error but error rates balloon when ability is low and task difficulty is high. Clinicians must keep cognitively vulnerable patients out of this territory because, like mental quicksand, it sucks them into nonadherence.

## 5. Diabetes Exemplifies How NCD Self-Management Regimens Invite Patient Error and Nonadherence

Poorly controlled diabetes elevates blood glucose (BG) levels. Persistently too-high BG causes blood cell clumping and increases blood viscosity, which impairs blood flow. Poorly controlled diabetes disrupts the metabolism of fats too, so adds excess lipids and cholesterol to circulate with the excess glucose. This toxic sludge slows healing and progressively damages organ systems body-wide, eventuating in one or more of diabetes’ complications, such as limb amputation and retinopathy. Diabetes is the leading cause of cardiovascular disease, end-stage kidney disease, and, among 18–64-year-olds, new cases of blindness in the U.S., virtually all preventable ([11]). It now accounts for one in four healthcare dollars in the U.S. ([2]), partly due to high rates of preventable emergency department (ED) use and hospitalization for dangerous BG levels ([17]).

Diabetes is an especially taxing NCD to self-manage, as the job description in Table 3 suggests. Patients must take hands-on control of a metabolism no longer on auto-pilot to keep BG levels within a target range, neither too high nor (if using insulin) too low. This requires simultaneously coordinating three inputs that affect BG level (medication, carbohydrate, physical activity) and factoring in other circumstances as well (current BG, illness) to keep pre- and post-meal BG levels moving within the target range. 

Diet is important in all NCDs but especially tricky in diabetes because patients must calibrate carbohydrate intake. Nutrition labels are helpful, but even their simplest uses require considerable information processing. A task analysis of one such use in Appendix A illustrates how many inconspicuous elements of a task can raise its complexity. The sample items in Figure 3 suggest that even simple-seeming uses of nutrition labels are more cognitively demanding than level 1 or 2 literacy tasks. 

Other diabetes self-management (DSM) tasks are yet more cognitively demanding. For instance, providers often ask people who are newly diagnosed or having difficulty controlling their BG levels to record their BG readings, medication, and perhaps carbohydrate intake by meal or time of day in a multi-row, multi-column chart. This is at least a literacy level 3 task. They also advise patients to look for patterns in their BG readings relative to food, medication, and physical activity and to adjust one or more of them, as necessary, to improve BG control. Such analyses and judgments—do-it-yourself research—are clearly level 5 difficulty.

Moreover, DSM is not a series of disconnected tasks, as are literacy surveys. It is a job, though one with no days off and scant training: a constellation of tasks that must be prioritized and coordinated to meet a specified goal. The job description in Table 3 leaves no doubt that DSM sits atop the peak of the error landscape in Figure 3. Good judgment is especially critical when self-administering insulin, because using the wrong type or amount, failing to eat enough carbohydrate or soon enough after administration, and other miscalculations and misadventures can land one in the ED ([17]). In the U.S., insulin is second only to the blood thinner Warfarin in ED visits for adverse drug events ([39]). As of 2018, 32.2% of all adults with diagnosed diabetes in the U.S. were prescribed insulin to manage their BG, not just the 5% with type 1 who must use it to survive (CDC, https://gis.cdc.gov/grasp/diabetes/DiabetesAtlas.html (accessed on 5 December 2021)).

DSM becomes even more complicated and error prone when, as is typically the case, information is incomplete and conditions are ambiguous, changing, uncertain, unpredictable, or stressful. Effective BG control is never certain for even the most conscientious and capable adherents to DSM regimens. Anything that affects metabolism can disrupt control, often making out-of-range BG levels difficult to anticipate, explain, and prevent. The emotional and cognitive burden can skyrocket when family and work schedules change, mealtimes are irregular, carbohydrate content of meals is unknown or highly variable, and life’s ups and downs divert time and attention (for examples, see [34], [35], [36], [37]; [38]). Not only do delayed meals, unanticipated surfeits or deficits of carbohydrate, night shifts, and stress affect metabolism directly, but their sometimes unpredictable effects on BG demand more analysis by patients (why did that happen?) and decision making (what do I do now?). 

By any measure, DSM is a complex job, as complex as many mid-level occupations ([19]). It is also a fast growing one. Diabetes cases quadrupled worldwide in the last four decades, reaching a global prevalence of 8.5% among adults by 2014 (https://www.who.int/news-room/fact-sheets/detail/diabetes (accessed on 5 December 2021)). 

## 6. New Hope: The Medical Professions Could Adapt Existing Person-Job Match Tools and Techniques to Help Clinicians Increase the Cognitive Accessibility of NCD Self-Management

Our best hope for meeting individual patients’ needs while controlling the crushing costs of nonadherence may lie with improving patients’ cognitive access to NCD-SM. Scott Adam’s “amazingly smart deviants” could help by developing the tools listed in Table 4 and redesigning electronic medical records systems to both enable and incentivize their use. The medical profession would also benefit from healthcare quality metrics and decision trees for individualizing self-care regimens to accommodate differences in cognitive capacity, just as there are for selecting appropriate medical treatments for patients with different medical profiles.

Medical and health associations regularly update their evidence-based standards of care and practice guidelines on patient care. All urge clinicians to consider many patient attributes, but rarely do they mention cognitive capacity. They always urge taking health literacy into account, but are vague about whether it is a general capacity for processing information or a teachable set of specific KSAs. They also urge screening older individuals for dementia, as appropriate, but otherwise imply that cognitive capacity is dichotomous—normal or abnormal. Care standards have begun to identify regimen complexity as a barrier to adherence for older adults, but seldom mention it otherwise.

Fortunately, experts in job analysis, personnel selection, job performance, psychometrics, and allied fields have developed tools and techniques for other cultural institutions that must take in, assess, place, and train individuals whose wide differences in cognitive capacity affect how quickly and well they learn a curriculum or job: government-supported schools, large corporations and government agencies, and the military services. All must attempt person-job match, especially cognitive match, to optimize individual-level and/or institutional-level performance. Many of these experts belong to the American Psychological Association’s Division 14 (Industrial and Organizational Psychology).

Health organizations, agencies, and insurers could engage these experts to adapt existing tools and techniques to reduce patient nonadherence. Table 4 (Part A) lists ones that would assist healthcare providers. Part B offers strategies for how clinicians might use them. These tools can optimize patient mastery and outcomes of NCD-SM, but few healthcare providers have the wherewithal to develop them on their own.

## 7. Additional Hope: Public Health Researchers Could Estimate the Global Disease Burden Attributable to Cognitive Factors and Identify WHO “Best Buys” for Reducing It

Epidemiologists have already done monumental work identifying trends in risk factors and best buy interventions for improving global health. Hundreds, if not thousands, have banded together in consortia, such as the NCD Risk Factor Collaboration (NCD-RisC) and the Global Burdens of Disease (GBD) Study, to scour the world for evidence and model it to identify successes, troubling trends, and ways to meet unmet needs. They could accelerate their contributions by adding two health risks to their roster: cognitive capacity of persons and complexity of task environments.

Population exposure to cognitive risk (the horizontal axis in Figure 3) can be estimated with publicly available data from professionally developed international assessments of adult literacy (Figure 2) and school learning (e.g., PIRLS, Progress in International Reading Literacy Study, https://www.iea.nl (accessed on 5 December 2021), and TIMSS, Trends in International Mathematics and Science Study, https://www.iea.nl (accessed on 5 December 2021)). These results can also help identify global or national pockets of elevated cognitive risk. The importance of exposure to a health risk is estimated with risk-outcome pairs (how big an impact particular risks have on particular outcomes). The literatures on health literacy, intelligence, and cognitive epidemiology ([13]) are replete with risk-outcome pairs linking cognitive capacity and health outcomes at every stage of the NCD disease process in Table 1.

Population-level exposure to cognitive risk changes little over time and generations, so it is effectively a fixed constraint on person-job matching in a population.[note 2] But the other half of the match is not, as detailed in Table 4. Much like air pollution, the cognitive complexities in NCD prevention and self-management (the complexity axis in Figure 3) are modifiable environmental risks. Risk-outcome pairs (akin to the error probabilities in Figure 3’s vertical axis) are often used to identify the biggest drivers of the rising global burden of NCDs as well as opportunities for slowing or reversing it. Charting the landscape of cognitive-attributable errors and nonadherence to treatment can, in like manner, point to populations, places, and practices in NCD treatment where cognitive accessibility needs improving. Epidemiologists could, for example, estimate how the global burden of LYDs might change if interventions decreased certain types of nonadherence, especially among individuals and families with high exposure to elevated cognitive risk. Only with such information can we avert the “slow-motion disaster” that NCDs portend.

## Figures and Tables

**Figure 1 jintelligence-09-00061-f001:**
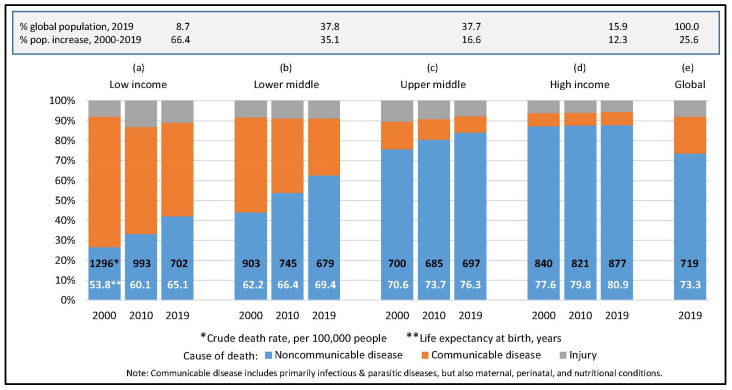
Trends in death rate, cause of death, and longevity at birth, by World Bank country income group, 2000–2019; All data available or calculated from data in the WHO’s online GHO database. For longevity: https://www.who.int/data/gho/data/indicators/indicator-details/GHO/life-expectancy-at-birth-(years). For all else: https://www.who.int/data/gho/data/themes/mortality-and-global-health-estimates/ghe-leading-causes-of-death (both accessed on 5 December 2021).

**Figure 2 jintelligence-09-00061-f002:**
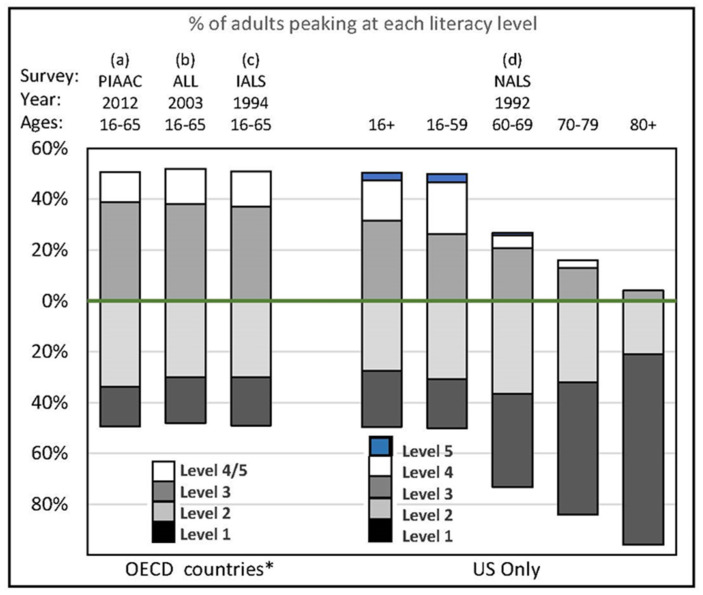
A nation’s distribution of adult functional literacy (information-processing capacity) is predictable. * N of countries in the OECD surveys, respectively = 20, 6, 8 (U.S. included in the PIAAC and IALS). PIAAC = Programme for the International Assessment of Adult Competencies, ALL = Adult Literacy and Life Skills Survey, IALS = International Adult Literacy Survey, NALS = National Assessment of Adult Literacy, OECD = Organisation for Economic Cooperation and Development. Sources for PIACC ([33], Table A2.1); ALL ([32]) and IALS ([31]) data from online database at https://piaacdataexplorer.oecd.org/ide/idepiaac (accessed on 5 December 2021); and NALS ([25], Figure 1.1 for ages 16+ and [7], Tables 1.2 and 1.3, for specific age groups).

**Figure 3 jintelligence-09-00061-f003:**
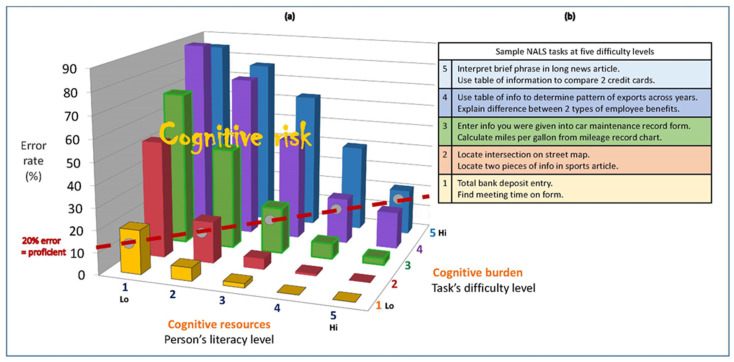
Sample literacy items and landscape of cognitive risk for individuals of lower literacy performing tasks of increasing difficulty. Sources for Panel (**a**), [24] ([24], Exhibit 13–36 for document scale); for Panel (**b**), [25] ([25], Figure 1).

**Table 1 jintelligence-09-00061-t001:** Percent (%) of global deaths in 2019 attributable to the 4 behavioral and 4 metabolic risk factors responsible for the global epidemic and burdens of NCDs.

Disease Process in Noncommunicable Diseases (NCDs)
Behavioral risks *(% of deaths caused)	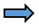	Metabolic risks *(% of deaths caused)	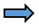	Top NCD causes of death **(% of all global deaths)	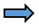	Health burdens
Tobacco use (15.4)		High systolic BP (19.1)		Coronary heart disease & stroke (27.8)		Comorbidities
Unhealthy diet (14.1)		High fasting BG (11.5)		COPD (5.8)		Hospitalizations
Alcohol misuse (4.3)		High BMI (8.9)		Tracheal, bronchus, lung cancer (3.6)		Years of disability
Sedentary (1.4)		High LDL cholesterol (7.8)		Diabetes (2.7)		Premature death
		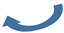		

* GBD level 2 risks, ** GBD level 3 causes. BG = blood glucose; BMI = body mass index; BP = blood pressure; COPD = chronic obstructive pulmonary disease; LDL = low-density lipoprotein. Source: [15] ([15]), with calculations on supplementary data at http://www.healthdata.org/results/gbd_summaries/2019 (accessed on 5 December 2021).

**Table 2 jintelligence-09-00061-t002:** Global burden of disease in 2019 by different metrics for the three broad categories of death (GBD level 1 causes) and the five NCDs with the highest death rates (GBD level 2 causes).

Disease Categories	Cases, 2019		Global Age-Adjusted Rate per 100,000 Persons, 2019					
	Most Ages	Millions	Prevalence	Incidence	Deaths	YLLs	YLDs	DALYs
Injuries	Teen-mid	1830	22,588	9259	55	2379	790	3169
Communicable diseases	Children	4540	58,287	346,347	141	8106	1377	9483
Noncommunicable diseases	Mid-late	7100	91,081	168,397	540	11,598	8607	20,205
Ischaemic (coronary) heart disease		197	2421	262	118	2177	67	2244
Stroke		101	1240	151	84	1550	218	1768
Chronic obstructive pulmonary disease		212	2638	201	43	681	245	926
Tracheal, bronchus, & lung cancer		3	39	27	25	545	7	552
Diabetes (both type 1 and type 2)		460	5555	268	20	416	443	859

YYL = years of life lost to premature death; YLD = years lived with a disability; DALY = disability-adjusted life years (YYL + YLD). Source: Supplemental 2-page summaries for [15] ([15]) at https://www.thelancet.com/gbd/summaries (accessed on 5 December 2021).

**Table 3 jintelligence-09-00061-t003:** The higher-order cognitive processing required for optimal diabetes self-management (DSM). (From “Safe-Guarding Cognitive Access to Diabetes Self-Management as Abilities Decline with Age” by [21]. Copyright 2021 by Canadian Diabetes Association).

Job of DSM
**Purpose:** Keep diabetes under daily control in the often changing and unpredictable circumstances of everyday life. **Goals:** Near term: Keep blood glucose (BG) within normal limits. Long term: Avoid complications and maintain quality of life. **Major duties:** Coordinate activities that influence BG (food, medication, physical activity). Anticipate effects on BG of those activities and their relative timing. Recognize symptoms indicating that BG is too low or too high. Adjust food, medicine, physical activity (as needed) to maintain or regain optimal BG. Obtain BG data from glucose meter or continuous glucose monitor to determine if BG is trending to hypo- or hyperglycemia. Determine timing and type of corrective action when BG levels are too low (glucose tablets, glucagon, emergency medical care). Detect and seek treatment for complications of elevated BG levels (vision changes, neuropathies, foot ulcers). Plan ahead for the unexpected and unpredictable (delayed meals, delayed or missed medication). Adjust DSM for other influences on BG (infection, emotional stress, insufficient or poor-quality sleep). Coordinate DSM with other self-care regimens (comorbidities, polypharmacy). Manage conflicting demands on time and behavior (DSM, family, work). Update DSM skills and knowledge, as needed (changes in technology, medication, impairments, comorbidities).

**Table 4 jintelligence-09-00061-t004:** Strategies to help clinicians reduce a patient’s cognitive barriers to NCD self-management.

Recommendations for increasing the cognitive accessibility of NCD self-management regimens	
**A. Medical/health associations & researchers develop new training, tools, & techniques for clinicians**	
*Persons: cognitive capacity & false beliefs*	
A.1	Add cognitive-access-to-care modules to medical and public health training programs. They would explain the wide variation in people’s cognitive needs and how to meet them. Physical, financial, and cultural access to NCD care mean little without cognitive access to it.
A.2	List the common misunderstandings and false beliefs that patients bring into care. Diabetes Disasters Averted (http://www.diabetesincontrol.com/resources/disasters-averted (accessed on 5 December 2021)) proves that nothing should be assumed too simple or obvious to need explaining.
*Jobs: cognitive complexity*	
A.3	Write job descriptions for all NCDs. Use both clinicians and patients or their caregivers as subject matter experts. The later will help care teams better conceptualize what patients have to manage and coordinate in real-world settings. See [38] ([38]) and [34] ([34], [35], [36], [37]) for compelling audio and written accounts by parents of type 1 children.
A.4	Expunge needless complexity from written materials for patients (e.g., no jargon, no long contorted sentences, clear organization, informative headings). See the U.S. Centers for Disease Control’s Plain Language guides ([10]). They help whittle complexity down to what is inherent.
A.5	Audit the cognitive demands inherent in effective NCD self-management. Engage job analysts to identify the information-processing requirements in typical regimens, including their configuration of tasks.
A.6	Perform task analyses of the most critical tasks in self-managing a particular NCD and where patients are most vulnerable to error. See [24] ([24]) for research on what adds to a task’s complexity.
A.7	Compile a list of common errors in self-management so that practitioners can anticipate and preempt them. Search the literature and survey practitioners.
A.8	Compile a list of the most dangerous patient errors in NCD-SM. For diabetes, see studies of preventable ED visits and hospitalizations for hypo- or hyperglycemia ([17]). They reveal the sorts of seemingly obvious facts that patients may need to be explicitly (re)taught.
A.9	Identify self-care tasks that the average person is not likely to perform correctly unless they get extra instruction. Use the landscape of error in Figure 3.
**B. Clinicians iteratively adjust self-care regimen and training to fit a patient’s cognitive needs**	
*Person: cognitive needs, barriers, and resources*	
B.1	Screen for dementia, if suspected. There are no short, unobtrusive tests of cognitive capacity in the normal range (from the 2nd to 98th percentile), nor is one needed. The patient’s performance on the criterion--self-management—is the best guide to next steps in adjusting their NCD-SM tasks and training. See B.7-15 below.
B.2	Determine whether the patient has functional impairments (e.g., sight, hearing, touch, swallowing) or comorbidities. All make NCD-SM more difficult and error-prone, the latter by multiplying the NCD-SM tasks, medications, and doctors a patient must coordinate.
B.3	Elicit the patient’s questions, concerns, and beliefs about their NCD and NCD-SM. False beliefs must be preemptively corrected lest they impede NCD-SM. Patient questions and concerns indicate not just the patient’s particular needs and preferences for regimen content, but also their knowledge and intellectual skills for implementing the regimen.
B.4	Be aware, however, that patient reporting is also a cognitive exercise. For instance, the patient may not know what is relevant. Older adults are especially reluctant to reveal declining mental capacity, but see [20] ([20]) for interview questions to elicit their cognitive needs and capacities.
B.5	Identify sources of cognitive support and interference in NCD-SM. Informal sources of information or support can be badly mistaken (e.g., friends offering leftover insulin). Knowledgeable family members can be valuable partners in NCD-SM.
B.6	Identify situational disruptions to self-management. Keeping external circumstances under better control can help patients keep blood glucose under better control. Routine is an underappreciated tool for the diabetes toolkit that many patients carry everywhere.
*Person-job cognitive fit: the regimen*	
B.7	Estimate a conservative starting point for a regimen’s complexity. For this, use any tools available from Section A above, patient attributes in B.1-6, and the landscape of error in Figure 3. Complexity can be increased over time once patients experience some success. Nothing builds self-confidence and motivation as well as developing actual competence.
B.8	Monitor patient difficulties and errors at successive levels of task difficulty. Locating their errors in the matrix of error probabilities (Figure 3) reveals where cognitive demands must be lightened to avoid pushing the individual into cognitive overload.
B.9	Administer a diabetes distress scale or equivalent to identify possible sources of cognitive overload. Remediate overload before assuming that a patient needs treatment for its natural sequelae: depression, anxiety, and loss of motivation.
B.10	Simplify regimens when necessary to bring them back within the individual’s cognitive reach. No matter how few self-care tasks a patient eventually masters, each one mastered does far more good than them giving up altogether.
B.11	Enlist cognitive assistance from capable caregivers or qualified health care providers if the individual cannot safely self-manage their NCD.
*Person-job cognitive fit: the training for it*	
B.12	Sequence instruction for efficient learning. Teaching tasks in order of their information processing complexity eliminates the needless cognitive hurdles that poorly organized instruction so often imposes on learners. The classic tool for this in school settings is Bloom’s taxonomy of cognitive educational objectives, from least to most cognitively complex ([3]). Appendix A lists typical components of DSM ordered by Bloom level. This tool doesn’t eliminate the inherent information processing demands it helps to reveal, but helps ensure that individuals grasp a task’s prerequisites before attempting it.
B.13	Adjust learning demands up or down in complexity to identify the individual’s “desirable difficulty range” for learning ([27]). This is like computer adaptive testing, where the first items administered are of middling difficulty but subsequent items increase or decrease in difficulty depending on the individual’s errors on prior items.
B.14	Adjust the pace, depth, breadth, and abstractness of material taught to fit the individual’s ability to take it in. Low ability learners benefit most from highly structured, detailed, concrete, contextualized, hands-on, theory-free, step-by-step instruction of task-specific skills. High ability learners benefit most from the opposite: abstract, theoretical, self-directed, and incomplete instruction that frees them to organize new and old information in novel ways ([26]). Slower instruction necessarily means covering less content.
B.15	Triage instructional content as necessary. Winnow SM tasks first by how critical each is to the patient’s well-being but exclude those too hazardous for that patient to attempt.

## Data Availability

All data in this article came from publicly available online databases and reports. Data in the figures and tables came from these four institutional sources: the Global Burden of Disease Project (http://www.healthdata.org/gbd/about (accessed on 5 December 2021)); the Organisation for Economic Cooperation and Development (OECD, https://www.oecd-ilibrary.org (accessed 5 December 2021)); the U.S. Department of Education’s National Center for Education Statistics (NCES, http://nces.ed.gov (accessed on 5 December 2021)); and the World Health Organization’s (WHO) Global Health Observatory (GHO) database (https://www.who.int/data/gho (accessed 5 December 2021)). Links for specific data are provided in either the text or the references.

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
