# Peer review of "The Transition to Noncommunicable Disease: How to Reduce Its Unsustainable Global Burden by Increasing Cognitive Access to Health Self-Management"

_jintelligence, 2021, doi:10.3390/jintelligence9040061_

Round 1
Reviewer 1 Report
The paper reviews cognitive aspects of handling one’s diabetes mellitus (DM). It is a thorough review, but the conceptual framework and the main argument are ambiguous for several reasons.
- The first half of the paper reviews noncommunicable diseases (NCD) in general, but the second part of the paper, when discussing the role of cognitive ability, focuses on DM almost exclusively. While the argument regarding the cognitive load of managing one’s blood glucose levels as the joint function of CH intake, medication and activity seems compelling, it is far less evident why the same or even similar level of self-management of cognitive load would be relevant for COPD, for instance. In fact, the paper comes across as the merging of two separate papers: a general review of NCD and one of cognitive abilities & DM.
- Relatedly, since the actual focus of the paper is DM, its title is somewhat misleading. The title should emphasise DM not NCD in general.
- It is unclear why g-theory is strongly relevant for the paper’s argument. See, for instance.: “8. g Theory Gives New Hope for Controlling the Unsustainable Costs of Global Epidemics of Noncommunicable Disease” and “Our best hope for meeting individual patients’ self-care needs while controlling the crushing costs of nonadherence may lie with g theory.” The paper’s main message is that one’s intellect has important implications’s for one’s ability to manage BG in DM. g-theory is unnecessary for an emphasis on cognitive abilities. That is, g-theory, in most of the literature, refers to interpreting the general factor of intelligence (g) as general cognitive ability. g-theory is in fact controversial, and challenged, for instance, by mutualism or process overlap theory, which explain the general factor without assuming a general cognitive ability and which therefore interpret g as some kind of mental index composed of several faculties. Not only is g-theory controversial, this controversy about the theoretical status of g is completely irrelevant from the perspective of the paper. The important thing for the argument is that cognitive abilities really matter - regardless of whether g reflects a single ability or whether it is a composite of several abilities. Therefore it seems unnecessary, and also somewhat misleading, to bring up g-theory the way it appears in the paper.
- “Small seemingly inconsequential cognitive errors, if consistent, cumulate over time or populations into big effects on outcomes (Spearman-Brown Prophecy Formula in psychometrics).” I am puzzled as to why the Spearman-Brown Prophecy Formula is mentioned here. This formula provides the extent of which the reliability (internal consistency) of a test changes as the function of the number of items, provided that certain assumptions are met. I cannot see the relevance. Please explain.
- It is argued that literacy is a good proxy for information processing and, ultimately, g. This is the basis of the argument that, even though what is discussed is in fact literacy, the ultimate causal factor is g. I see two problems with this:
- Even though literacy is a good proxy for g in wester countries, the paper also deals with non-western, third world countries, where variation in literacy might reflect differences in access to formal schooling, too, a factor less important in the West.
- Specific skills more directly related to BG management might have a more proximal effect than g. This is in fact acknowledged by the author, e.g. “g is an especially well-validated psychological construct (…) However, a different construct is required to understand how g actually has its effects.” and, importantly: “The provider need not assess patient capacity level. Instead, they need to ascertain an individual’s difficulties and errors, including self-reported, when they attempt learning and doing tasks at successive levels of task difficulty. Locating their errors in the matrix of error probabilities reveals where cognitive demands must be lightened to avoid pushing the individual into cognitive overload.” It is unclear, therefore, why the focus of the paper is on g, when this is not a construct that can be the target of any direct intervention, as pointed out by the author, too.
Overall, the paper is interesting and points to an important aspect of applied cognitive psychology. But a re-arrangement of its structure and a reconsideration of several lines of the argument in order to be more compelling and coherent would greatly improve it.
Author Response
Dear Reviewer 1,
Thank you for your detailed comments. They were very helpful.
Linda
Review 1
The paper reviews cognitive aspects of handling one’s diabetes mellitus (DM). It is a thorough review, but the conceptual framework and the main argument are ambiguous for several reasons.
The first half of the paper reviews noncommunicable diseases (NCD) in general, but the second part of the paper, when discussing the role of cognitive ability, focuses on DM almost exclusively. While the argument regarding the cognitive load of managing one’s blood glucose levels as the joint function of CH intake, medication and activity seems compelling, it is far less evident why the same or even similar level of self-management of cognitive load would be relevant for COPD, for instance. In fact, the paper comes across as the merging of two separate papers: a general review of NCD and one of cognitive abilities & DM.
Diabetes is the most complex of the major NCDs, but preventing and managing all of them is cognitive demanding. I’ve tried to make that point clearer by emphasizing that all reflect the same disease process. Exposure to known risk factors is voluntary: all are modifiable behaviors. Avoiding their predictable ravages requires looking past their immediate but unhealthy pleasures and conveniences to imagine complications far in the future—delay discounting, which is correlated with intelligence. In any case, people who get one chronic disease often get others as well (which I mention), because all are caused by the same sorts of internal damage caused by the same unhealthy behaviors (I am more explicit about that now).
As I also mention, diabetes is the major cause of cardiovascular disease, which means that lots of people are managing both that and diabetes.
Relatedly, since the actual focus of the paper is DM, its title is somewhat misleading. The title should emphasise DM not NCD in general.
I hope that the focus on diabetes makes more sense in my revision. I used diabetes because it was a clear example of a cognitively demanding NCD, it is so common and increasing quickly everywhere, and is a major risk factor for other NCDs (because of its effect on blood sugar and lipids. My suggestions for improving cognitive accessibility (the added Table 4) apply to NCDs in general.
In fact, all discussions of the burdens of NCDs come back to diabetes, as I now explicitly indicate. The BGD papers I cite, as does the WHO, discuss how diabetes and obesity (its chief cause) are the key drivers of the NCD epidemic, especially because they disable much for so long—and cost so much-- before they kill, often with a heart attack.
1. It is unclear why g-theory is strongly relevant for the paper’s argument. See, for instance.: “8. g Theory Gives New Hope for Controlling the Unsustainable Costs of Global Epidemics of Noncommunicable Disease” and “Our best hope for meeting individual patients’ self-care needs while controlling the crushing costs of nonadherence may lie with g theory.” The paper’s main message is that one’s intellect has important implications’s for one’s ability to manage BG in DM. g-theory is unnecessary for an emphasis on cognitive abilities. That is, g-theory, in most of the literature, refers to interpreting the general factor of intelligence (g) as general cognitive ability. g-theory is in fact controversial, and challenged, for instance, by mutualism or process overlap theory, which explain the general factor without assuming a general cognitive ability and which therefore interpret g as some kind of mental index composed of several faculties. Not only is g-theory controversial, this controversy about the theoretical status of g is completely irrelevant from the perspective of the paper. The important thing for the argument is that cognitive abilities really matter - regardless of whether g reflects a single ability or whether it is a composite of several abilities. Therefore it seems unnecessary, and also somewhat misleading, to bring up g-theory the way it appears in the paper.
I agree. All that is gone. Intelligence researchers disagree about the nature of intelligence but, with few exceptions, accept a general cognitive ability factor as a valid behavioral phenotype. That’s what I now emphasize—the importance of the phenotype (not the agreement)
2. “Small seemingly inconsequential cognitive errors, if consistent, cumulate over time or populations into big effects on outcomes (Spearman-Brown Prophecy Formula in psychometrics).” I am puzzled as to why the Spearman-Brown Prophecy Formula is mentioned here. This formula provides the extent of which the reliability (internal consistency) of a test changes as the function of the number of items, provided that certain assumptions are met. I cannot see the relevance. Please explain.
I deleted that reference and just made the substantive point. Though I don’t put it this way in my revision, small consistent effects—like interest—compound over time, and certainly at the population level.
3. It is argued that literacy is a good proxy for information processing and, ultimately, g. This is the basis of the argument that, even though what is discussed is in fact literacy, the ultimate causal factor is g. I see two problems with this:
My reading of the evidence is that they all—g, literacy, and information processing capacity—represent the same empirical phenomenon. But I need not prove that here to make my point: which is that criterion-referenced literacy proficiency scores show that people differ a lot in their capacity to perform cognitively demanding tasks.
It is important to bring in intelligence because it is a known causal factor in job and training performance. It is, in fact, the best single predictor of both. And NCD-SM regimens can also be conceptualized as jobs, as my paper describes.
That is important also because my concern here is with the complexity of NCD-SM jobs, not the nature of intelligence. We need not settle the latter to be concerned about the real-world impact of the former. I don’t think you are saying that, but it is easy for intelligence researchers, obviously including me too, to focus on what intelligence, the latent trait, is rather than on what in our environment calls that phenotype into action and thus makes it observable.
4. Even though literacy is a good proxy for g in western countries, the paper also deals with non-western, third world countries, where variation in literacy might reflect differences in access to formal schooling, too, a factor less important in the West.
It may very well be true that adult literacy surveys in third-world countries wouldn’t be as valid for distinguishing levels of information processing capacity as those in my Table 2, just as I would argue that a verbal intelligence test might not be valid either if it requires language and facts foreign to the person. But I don’t think that’s relevant here. First, the rather few adult literacy tests available are created in different languages and the content is everyday stuff familiar to the population, and vetted for that.
More to the point, the adult literacy data from high income countries in my Table 2 make the point that, even when schooling is universal, people differ greatly in general ability. (Universal access actually makes those differences more evident precisely because it is universal.) Those data also show in concrete terms how the latent capacity greatly affects our ability to negotiate the complexities of modern life. In fact, having universal schooling is one of the strengths of these literacy data because we can determine the functional value of higher intelligence (greater information processing capacity) without it being confounded by lack of access to schooling.
Capacity and knowledge may not be as tightly correlated when schooling is not universal, but that does not change the fact that fluid intelligence (which coincides with g itself in many studies) differs greatly within all human populations. Differences in access to schooling only changes who accumulates relatively more school-taught knowledge and skills. Although lack of universal schooling might lower the correlation between intelligence and knowledge, that would not mean that schooling is more important than intelligence. In any case, general intelligence is fungible but knowledge is not. This is not to say that knowledge doesn’t matter. It means only that differences in cognitive capacity continue to matter a lot regardless of one’s level of schooling or knowledge. The job performance literature make that crystal clear.
There are several excellent international tests of school achievement (learning in school) that I now suggest as possible tools for getting at the distribution of cognitive ability in developing countries. But my concern is not to assess cognitive capacity, but to figure out how to accommodate the big differences that will exist everywhere, schooled or not. See if my paper is clearer on that now.
5. Specific skills more directly related to BG management might have a more proximal effect than g. This is in fact acknowledged by the author, e.g. “g is an especially well-validated psychological construct (…) However, a different construct is required to understand how g actually has its effects.” and, importantly: “The provider need not assess patient capacity level. Instead, they need to ascertain an individual’s difficulties and errors, including self-reported, when they attempt learning and doing tasks at successive levels of task difficulty. Locating their errors in the matrix of error probabilities reveals where cognitive demands must be lightened to avoid pushing the individual into cognitive overload.” It is unclear, therefore, why the focus of the paper is on g, when this is not a construct that can be the target of any direct intervention, as pointed out by the author, too.
The construct in question is not intelligence, but the criterion performance. I hope that is clearer now.
My revision clarifies that g is not the focus of the paper, but the quality of NCD-SM—as you indicate. The direct target of intervention, however, is not the patient but the patient’s NCD-SM regimen and how it is taught. They must be tailored to fit in the individual’s capacity to optimize their learning and job performance. So, level of g/literacy is the constraint, jobs/tasks are the variables to be manipulated, and quality of self-care is the criterion.
Performance, as on any job, requires subject matter knowledge and skills, which is the purpose of diabetes and other NCD education. But all jobs also require exercising judgment on when and how to use that declarative and procedural knowledge to meet a specified goal. They may be necessary but are never sufficient. Moreover, they can become outdated quickly, so the individual must continually update their knowledge and skills, usually on their own. That is what job analysis research shows. In any case, people with diabetes are generally taught only the barest amount of knowledge and skills to control their condition. Hint: it’s not reimbursable, for the most part.
Even if there were properly trained, the job performance literature shows that nobody can ever be trained completely for any job because all require some information gathering and problem solving. Diabetes requires a lot of that, more than many paid jobs. That is why the notion of health literacy as just a collection of skills and knowledge is wrong-headed. Such collections can never substitute, at least for long, for the capacity to reason and solve novel problems (once again, as the job performance literature shows). As the functional literacy surveys also indicate, it is the ability to acquire and use that information that matters most in forecasting performance on common everyday tasks. I have scant space to talk about that, but I’ve tried to clarify it.
Though my paper does not mention it, it’s disconcerting how little education and support patients with diabetes get. Being told what the job requires and how to do it certainly is important! That’s why it’s so sad that people with diabetes typically get so little training and feedback on how to manage their condition. I don’t think most prescribing doctors even know what it takes unless they themselves or their children have it. You might read the Seitles' articles I cite for insight on what it means to be left without the specific skills and knowledge you so desperately need. It shouldn’t have to be a learn-as-you-stumble-along job.
Overall, the paper is interesting and points to an important aspect of applied cognitive psychology. But a re-arrangement of its structure and a reconsideration of several lines of the argument in order to be more compelling and coherent would greatly improve it.
Thank you. I hope my condensed and more tightly argued revision does, in fact, accomplish this. I would welcome you letting me know if, and where, it does not.
Linda
Submission Date
03 August 2021
Date of this review
22 Sep 2021 11:27:21
Reviewer 2 Report
The Transition to Noncommunicable Disease: How g Theory Can Reduce Its Unsustainable Global Burden by Increasing Cognitive Access to Health Self-Management
General comment. My opinion is that the paper is much too long, and that most of the sections could be stated far more briefly. Given that the argument is that intelligence/health literacy might be important for complex health care, it seemed to be odd—and a very important omission—that the author did not tell the reader about the many studies that report associations between intelligence/health literacy and the 18 cells in Table 1; as far as I am aware, there are studies relating intelligence to all of these—surely that would be a major evidential plank of the paper, in terms of convincing the reader that the central case was valid. Another general comment is that there was no indication of the variance accounted for/population attributable fraction or other such effect size that exists with respect to the exposures and outcomes mentioned. Many of the associations that the author refers to or hints at—either between elements of Table 1, and/or between intelligence/health literacy and elements of table 1—are quite small, though maybe important at the population level; however, the author should, on the basis of the actual studies, indicate what might be achieved at the population level, and what the associations might mean for an individual and, perhaps, change in behavior.
Section 2: Given that this is not as medical journal, the main non-communicable diseases should be listed early on. What is the evidence that most cases of con-communicable disease are preventable? When the author writes that “they result from common everyday habits” what is the accounted-for proportion of cases based on the habits (which should be listed)?
Section 3. I was wondering if most of this was needed. The point could be made and evidenced briefly. The article’s point is to show how cognitive ability might relate to non-communicable disease, not to describe the transition of disease types.
Section 4: I thought that this was much too long; I do not think all the text or the Figures are needed. Also, it omitted to include/state clearly/highlight some of the more important parts of evidence that would be needed to make the author’s case. That is, based on Table 1, one wants to be given evidence for the accounted-for variance in these illnesses based on both the “unhealthy behaviours” and “physiological harm”. I am not concerned that this be expressed in any one way. However, the ‘population attributable fraction’ (PAF; see https://www.bmj.com/content/360/bmj.k757 ) is used by epidemiologists and would be appropriate. With regard to diabetes, should the author specify Type 2 Diabetes? Also, there are other smoking-associated cancers than lung cancer. Which of the illnesses is caused by alcohol consumption?; it has an odd association with health, i.e. it is not always harmful in association studies.
Section 5. Again, this seemed over-long given the point that was being made, and some of the results one would expect to see was not summarised. There are associations out-there between intelligence and health literacy (more than just the literary association that the author refers to in Gottfredson, 1997) and intelligence, and health literacy and illnesses. Overall, I found there was too much in principle/rational argument but too little citation of relevant empirical evidence.
Section 6. This was overly long. It could have been summarised in a paragraph or so. As above, most of the cognitive difficulty case seemed to be based on in-principle argument, whereas one wants to be shown what proportion of the problem is accounted for by health literacy/intelligence.
Section 7. This seemed to be much too long. It is called ‘empirical generalisations’ but there were few references. I thought the ideas here could be summarised in a fraction of the space.
Section 8. This is not what I was expecting at this stage; I had thought one might be told what the empirical associations are between cognitive ability and health literacy and between these two measures and the four columns in Table 1. At this point, the paper seemed to be Hamlet without the Prince; there are many empirical studies from the last 20-or-so years relating intelligence/health literacy to each other and to almost all of the items listed under ‘unhealthy behaviors’, ‘physiological harm’, ‘chronic disease’ and ‘outcomes’. The author writes: “Our best hope for meeting individual patients’ self-care needs while controlling the crushing costs of nonadherence may lie with g theory.” The studies the conduct associations that could support that have been done, are many in number, and are not cited or described; this seems strange. Why make this very long, largely in-principle case for the importance of intelligence/health literacy in chronic/complex diseases when there are lots of studies reporting associations already out there?
Author Response
Dear Reviewer 2,
Thank you for your helpful comments. I have greatly shortened the paper, tightened its line of argument, added PAFs, and addressed other concerns you had in the process.
Linda
Review 2
General comment. My opinion is that the paper is much too long, and that most of the sections could be stated far more briefly.
I agree.
Given that the argument is that intelligence/health literacy might be important for complex health care, it seemed to be odd—and a very important omission—that the author did not tell the reader about the many studies that report associations between intelligence/health literacy and the 18 cells in Table 1; as far as I am aware, there are studies relating intelligence to all of these—surely that would be a major evidential plank of the paper, in terms of convincing the reader that the central case was valid.
These relations are indeed well established. I now urge epidemiologists to use those risk-outcome data when they model the attributable fractions of different risks for different health outcomes.
The main point of my paper was not to argue that intelligence is important, but to understand why it is. We need to know why literacy and general intelligence are so important in health matters if we are to help people of lower or declining intelligence. That requires looking at what in the work of self-care itself calls for people to process more information and hence make their intelligence levels more salient. Mine is a quite practical aim.
Another general comment is that there was no indication of the variance accounted for/population attributable fraction or other such effect size that exists with respect to the exposures and outcomes mentioned. Many of the associations that the author refers to or hints at—either between elements of Table 1, and/or between intelligence/health literacy and elements of table 1—are quite small, though maybe important at the population level; however, the author should, on the basis of the actual studies, indicate what might be achieved at the population level, and what the associations might mean for an individual and, perhaps, change in behavior.
My argument does not rest on high individual-level correlations with intelligence/literacy or percents of variance in health outcomes accounted for. Nor, as you would know, do they need to be large to have a big effect at the population level. Epidemiologists work at a different level of aggregation than do psychologists.
But you are correct that PAFs would help, so I now provide some, mostly for the known behavioral and metabolic risks responsible for so much NCD mortality. Thanks for the nudge.
I don’t give any for intelligence, because I know of none. But I urge epidemiologists—many publishing in the Lancet and contributing their data to the online WHO database--to start producing them. I suggest two cognitive risks they should add to their nearly 100: levels of cognitive capacity in a population and levels of cognitive complexity in their environments. That could help answer your/our big question: what difference would it make to increase the cognitive accessibility of NCD self-management? Health literacy researchers call for it too, but have not quantified its possible benefits—nor do I think they have the necessary analytic skills to do so.
Section 2: Given that this is not as medical journal, the main non-communicable diseases should be listed early on. What is the evidence that most cases of con-communicable disease are preventable? When the author writes that “they result from common everyday habits” what is the accounted-for proportion of cases based on the habits (which should be listed)?
All the big health agencies, WHO, CDC, etc, say they are preventable. The main evidence in my view for practically all NCDs being preventable is that all their major risk factors are modifiable—behaviors like eating to excess, smoking, etc. I continue to cite a recent report on NCDs from the American Heart Association, which reviews evidence that these behaviors are known risk factors for all the major NCDs, which I list. And, as I mentioned above, I have given attributable fractions for the behavioral risks. They are astonishing, very worrying.
Section 3. I was wondering if most of this was needed. The point could be made and evidenced briefly. The article’s point is to show how cognitive ability might relate to non-communicable disease, not to describe the transition of disease types.
No, it was not. I confine the discussion now to showing how the disease process is so cognitive for noncommunicable diseases but not for infectious diseases. Epidemiologists describe an individual’s exposure to known risks for the former as voluntary, but as involuntary for the latter.
Section 4: I thought that this was much too long; I do not think all the text or the Figures are needed. Also, it omitted to include/state clearly/highlight some of the more important parts of evidence that would be needed to make the author’s case. That is, based on Table 1, one wants to be given evidence for the accounted-for variance in these illnesses based on both the “unhealthy behaviours” and “physiological harm”. I am not concerned that this be expressed in any one way. However, the ‘population attributable fraction’ (PAF; see https://www.bmj.com/content/360/bmj.k757 ) is used by epidemiologists and would be appropriate. With regard to diabetes, should the author specify Type 2 Diabetes?
I agree. I eliminated most of the figures and condensed the discussion of data to key points. See also my reply above to your suggestion for PAFs above.
I have clarified the difference in the disease process for type 1 and type 2 diabetes, partly because so many people with type 2 also use insulin, which is risky business.
Also, there are other smoking-associated cancers than lung cancer. Which of the illnesses is caused by alcohol consumption?; it has an odd association with health, i.e. it is not always harmful in association studies.
Interesting question, but not for this paper. The 2019 American Heart Association report I cite provides a nice review of how all the risks relate to all the major NCDs. COPD is one of the smoking diseases, for sure. But smoking and alcohol misuse apparently do various sorts of physiological damage so are risks for a variety of NCDs. As is obesity, of course. I summarize their effect on all NCDs with PAFs for all-cause mortality.
Section 5. Again, this seemed over-long given the point that was being made, and some of the results one would expect to see was not summarised. There are associations out-there between intelligence and health literacy (more than just the literary association that the author refers to in Gottfredson, 1997) and intelligence, and health literacy and illnesses. Overall, I found there was too much in principle/rational argument but too little citation of relevant empirical evidence.
I must not have been clear enough in this section, so I hope I have now sufficiently clarified it. Except for showing the population distribution of literacy/intelligence available in a population (Figure 2), it concentrates on literacy researchers’ analyses of tasks to understand how they bring forth differences in the performance we conceptualize as “intelligence.” What can people actually do at different levels of proficiency? IQ tests can’t tell us that, which is one reason many people argue they can’t possibly measure or predict anything of consequence. Their items are just silly, abstruse, and academic.
In this section I therefore focus on what makes some tasks more difficult (more g loaded) than others. This section is thus about task demands, not people’s abilities. We need to know that because we can manipulate the tasks we ask patients to do (and how well we train them) but not how bright or proficient at learning adults are.
Section 6. This was overly long. It could have been summarised in a paragraph or so. As above, most of the cognitive difficulty case seemed to be based on in-principle argument, whereas one wants to be shown what proportion of the problem is accounted for by health literacy/intelligence.
I disagree. My aim here is show what makes diabetes so cognitively demanding and hence invites cognitive error (nonadherence). My experience working in that field and giving workshops is that most healthcare providers have no clue that self-care even makes cognitive demands. Or that people really do differ in cognitive ability, or that it might affect their ability to do what may seem to providers easy and obvious. That’s why I need to provide specific examples and tie them to the literacy data. I am not writing just for intelligence researchers.
I show the cognitive demands of diabetes self-care partly by providing a job description of it. Conceptualizing NCD self-management as a job also links patient nonadherence to the huge literature on jobs, job training, and job performance. This is a more general model for understanding patient nonadherence. I now make that explicit. In Table 4 I list specific ways the medical professions can exploit that expertise, for example, to audit the cognitive demands of NCD self-care. None of the NCDs is easy because they all share one particularly high cognitive barrier: expecting patients to forego evolutionarily novel here-and-now pleasures that don’t cause any observable harm in the near-term so they can avoid far-distant health complications. And many people have more than one NCD by old age. If you get diabetes, you will probably get a cardiovascular disease as well.
Section 7. This seemed to be much too long. It is called ‘empirical generalisations’ but there were few references. I thought the ideas here could be summarised in a fraction of the space.
I deleted this section and replaced it with several more focused paragraphs and a table listing specific suggestions for evidence-based “cognitive accessibility” tools and techniques the medical professions could develop or commission for use by clinicians. I also added a short section of suggestions for epidemiologists who study the global burden of NCDs, one being that they include two cognitive risk variables to their PAF modeling. Also said they ought to incorporate risk-outcome pairs for intelligence and health from research in cognitive epidemiology.
Section 8. This is not what I was expecting at this stage; I had thought one might be told what the empirical associations are between cognitive ability and health literacy and between these two measures and the four columns in Table 1. At this point, the paper seemed to be Hamlet without the Prince; there are many empirical studies from the last 20-or-so years relating intelligence/health literacy to each other and to almost all of the items listed under ‘unhealthy behaviors’, ‘physiological harm’, ‘chronic disease’ and ‘outcomes’. The author writes: “Our best hope for meeting individual patients’ self-care needs while controlling the crushing costs of nonadherence may lie with g theory.” The studies the conduct associations that could support that have been done, are many in number, and are not cited or described; this seems strange. Why make this very long, largely in-principle case for the importance of intelligence/health literacy in chronic/complex diseases when there are lots of studies reporting associations already out there?
This is not a play about Hamlet, if you mean making a case that intelligence matters for NCD risk factors and outcomes. I’ve done that in other publications, as do Ian Deary and others in this symposium. I hope my revision now clearly shows that my play is about something else entirely. Namely, for both patients’ and global well-being, we need to make NCD self-care more cognitively accessible to patients—make their job and its training less cognitively demanding—especially for the most cognitively vulnerable because their error rates are so needlessly high.
I don’t invoke g theory any more, and didn’t need to before.
Thanks again for your thought-provoking comments.
Round 2
Reviewer 1 Report
Thank you, Linda, for so carefully attending to my comments. I have no further recommendations or questions remaining. Best wishes, Kristof
Reviewer 2 Report
I retain some of the reservations I had about the first version of the paper--though I recognise and appreciate the many changes that have been made, especially cuts and abbreviating--but I also recognise that authors should be allowed to make their case in their own ways. This is an interesting, rich, and useful paper.